# Optimal transport based dimensionality reduction

## Abstract

This paper investigates whether modeling image and text data as probability measures and applying optimal transport (OT)-based dimensionality reduction techniques leads to improved performance in downstream machine learning tasks. We compare OT-based neighbor embedding methods to their Euclidean counterparts across both classification and clustering tasks using benchmark datasets: MNIST, Fashion MNIST, Coil-20, Yale Face, and 20-Newsgroups. Our methodology involves computing distance matrices using Wasserstein or Euclidean metrics, applying dimensionality reduction techniques such as MDS, Isomap, t-SNE, and Laplacian eigenmaps, and evaluating performance using standard classifiers and clustering algorithms. Experimental results show that OT-based embeddings often yield better performance, although there is some variance in datasets with textures like Fashion MNIST. For all experiments, we perform a statistical hypothesis test to support the findings.

## 1 Introduction

The purpose of this paper is to test the hypothesis that treating image data as probability measures and using optimal transport-based dimensionality reduction algorithms achieves better supervised and unsupervised learning task performance compared to treating data as Euclidean vectors and using standard Euclidean dimensionality reduction algorithms.

Modeling the geometry of a data set remains a crucial task in machine learning. Oftentimes, data may naturally be represented as a vector, matrix, or tensor, in which case the standard Euclidean distance is ostensibly the natural metric on the data. However, it is readily apparent that the Euclidean metric may not always be a semantically meaningful notion of geometry for all data types. Indeed, a pixelwise interpolation between two images leads to artifacts. With this in view, there have been several works that take an alternative modeling approach and treat images as probability measures on $\mathbb{R}^2$, e.g., (Wang et al., 2010; Kolouri et al., 2017; Moosmüller & Cloninger, 2023). From there, it is natural to utilize the Wasserstein distance arising from optimal transport (Villani, 2008; Santambrogio, 2015). Many works have studied how the geometry of data in Wasserstein space can be modeled, e.g., structured submanifolds of Wasserstein space (Hamm et al., 2023; Carlen & Gangbo, 2003; Gangbo & McCann, 2000), how it can be utilized in dimensionality reduction (Wang et al., 2010; Hamm et al., 2022; Cloninger et al., 2025), or how it can improve standard machine learning tasks like classification (Khurana et al., 2023; Shifat-E-Rabbi et al., 2024; Rubaiyat et al., 2024; Mathews et al., 2019; Aldroubi et al., 2021).

One of the reasons for the studies above is that mapping images to a probability distribution on a 2-dimensional grid better maintains the spatial meaning of the pixels, which may be lost during vectorization. This remains true for gene expression networks (Mathews et al., 2019), and NLP data also lends itself to utilizing optimal transport in a natural way, e.g., (Kusner et al., 2015). Another reason is that, optimal transport is based on morphing one distribution into another in the most cost effective way, and therefore gives a sometimes more semantically meaningful distance measure between data.

### 1.1 Main Contribution

The main purpose of this paper is to give more experimental evidence to the hypothesis implicit in many prior works: that modeling data using Wasserstein geometry can lead to better task performance. To test

this hypothesis, we consider a pipeline of the form

$$\text{data} \longrightarrow \text{dimensionality reduction} \longrightarrow \text{ML task}$$

where we use several known linear and nonlinear dimensionality reduction algorithms (using Euclidean distances or Wasserstein distances between data), and the ML task is either classification or clustering. For classification, we use several well-known classifiers on the embedded data, and for clustering we use $k$-means and spectral clustering.

To test our hypothesis, we run multiple replicate experiments on various benchmark datasets (see Section 2) and compare the task performance (e.g., accuracy) using a two-sample $t$-test. This allows us to make precise statistical statements on whether the optimal transport-based approach improves task performance.

### 1.2 Review of optimal transport

Let $\mathcal{P}_2(\mathbb{R}^n)$, the space of probability measures on $\mathbb{R}^n$ with finite 2-nd moment $(\int_{\mathbb{R}^n} |x|^2 d\mu(x) < \infty)$. The quadratic Wasserstein distance is defined by

$$W_2(\mu, \nu) := \inf_{\pi \in \Gamma(\mu, \nu)} \left( \int_{\mathbb{R}^n \times \mathbb{R}^n} |x - y|^2 d\pi(x, y) \right)^{\frac{1}{2}}, \tag{1}$$

where $|\cdot|$ is the Euclidean distance on $\mathbb{R}^n$, and $\Gamma(\mu, \nu) := \{\gamma \in \mathcal{P}(\mathbb{R}^n \times \mathbb{R}^n) : \gamma(A \times \mathbb{R}^n) = \mu(A), \ \gamma(\mathbb{R}^n \times A) = \nu(A) \text{ for all } A \subset \mathbb{R}^n\}$ is the set of couplings, i.e., measures on the product space whose marginals are $\mu$ and $\nu$. The formulation in equation 1 is called the Kantorovich formulation of optimal transport, and always admits a solution. For discrete measures, one can exactly compute a coupling via a linear program (Peyré & Cuturi, 2019), but one can also use entropic regularization to approximate the distance (Cuturi, 2013).

### 1.3 Review of distance based dimensionality reduction

Dimensionality reduction is a general tool in which one seeks an embedding of high-dimensional data into a low-dimensional (typically Euclidean) space such that a given task is more achievable on the embedded data. Oftentimes, one seeks embeddings that preserve some geometric or cluster structures in the data. One reason these methods have found utility is that they can sometimes mitigate the computational cost of a learning algorithm, where the curse of dimensionality appears in the low-dimensional embedding space rather than the high-dimensional data space.

We focus primarily here on neighbor embeddings (in the terminology of (Meilă & Zhang, 2024)), in which one first forms a neighborhood graph on the data with edge weights often informed by a metric on the ambient data space. The embedding is then often computed from the spectrum of a metric distance matrix (as in multidimensional scaling (MDS) or Isomap) or something derived from it (like the graph Laplacian as in diffusion maps). We use the general term neighbor embeddings for those based on metric distances in some fashion, and focus our attention here on MDS (Mardia, 1979), Isomap (Tenenbaum et al., 2000), t-SNE (van der Maaten & Hinton, 2008), and Laplacian eigenmaps (Belkin & Niyogi, 2003).

More concretely, suppose we have a data set in a metric space $\{x_i\}_{i=1}^N \subset (X, d)$. We denote the squared distance matrix by $D \in \mathbb{R}^{N \times N}$ where $D_{ij} = d(x_i, x_j)^2$. The MDS embedding into $\mathbb{R}^n$ is given by the first $n$ eigenvalues and eigenvectors of $-\frac{1}{2}\mathbf{1}\mathbf{1}^\top D \mathbf{1}\mathbf{1}^\top$ where $\mathbf{1}$ is the $N$-dimensional all-ones vector. Isomap computes a neighbor graph (e.g., $k$-nearest neighbors or $\epsilon$-neighborhood) on the data, computes a graph shortest path metric $(D_G)_{i,j} = d_G(x_i, x_j)^2$, then uses the spectrum of $-\frac{1}{2}\mathbf{1}\mathbf{1}^\top D_G \mathbf{1}\mathbf{1}^\top$ (i.e., Isomap is MDS on the graph metric). Laplacian eigenmaps forms a neighborhood graph on the data, computes a graph Laplacian $L$ (see (Von Luxburg, 2007) for a discussion of various graph Laplacians), and then embeds using the first $n$ eigenvalues and eigenvectors of $L$. Lastly, $t$-distributed stochastic neighbor embedding ($t$-SNE) (van der Maaten & Hinton, 2008) assigns a Gaussian probability distribution over the high-dimensional data, initializes a random low-dimensional embedding to which it assigns a $t$-distribution. Then the Kullback–Leibler divergence between the high-dimensional Gaussian and low-dimensional $t$-distributions is minimized over the low-dimensional embedding. For more details see (van der Maaten & Hinton, 2008).

## 2  Datasets

We use a variety of data sets for the experiments to evaluate our results. For classification tasks, we use MNIST, Fashion MNIST, Coil-20, and 20 Newsgroup datasets.

**MNIST:**  MNIST (LeCun, 1998) is a widely used, now classical, dataset for machine learning, and consists of grayscale (one channel) images of handwritten digits from $0 - 9$. There are $70,000$ images from the 10 classes, each of which is represented as a $28 \times 28$ pixel array with integer values from 0 (black) to 255 (white). The pytorch version of MNIST splits the dataset into $60,000$ training images and $10,000$ test images.

**Fashion MNIST:**  The Fashion MNIST dataset (Xiao et al., 2017) is formatted similar to MNIST (70,000 grayscale images of resolution $28 \times 28$ split into 60,000 training images and 10,000 test images). This dataset also has 10 classes: T-shirt/top, Trouser, Pullover, Dress, Coat, Sandal, Shirt, Sneaker, Bag, Ankle boot.

**Coil-20:**  The COIL-20 dataset (Nene et al., 1996) is a rotational image dataset consisting of $128 \times 128$ color (RGB) images from 20 object classes, with images of each object captured from 72 evenly spaced viewing angles at 5-degree intervals, resulting in a total of $1,440$ images.

**20-Newsgroups:**  The 20-Newsgroups dataset (Lang, 1995) is a text document corpus of newsgroup documents from 20 classes, and is used in natural language processing (NLP). The classes contain broad categories such as recreational activities, science, computers, religion and politics. Each of these further divided into subcategories. Dataset comprised with approximately $20,000$ data points (as a text document) and roughly equally distributed among the classes (i.e. Each classes contains $1,000$ data points). The classes are: Comp.graphics, Comp.os.ms-windows.misc, Comp.sys.ibm.pc.hardware, Comp.sys.mac.hardware, Comp.windows.x.rec.autos, Rec.motorcycles, Rec.sport.baseball, Rec.sport.hockey, Sci.crypt, Sci.electronics, Sci.med, Sci.space, Misc.forsale, Talk.politics.misc, Talk.politics.guns, Talk.politics.mideast, Talk.religion.misc, Alt.atheism, and Soc.religion.christian.

**Yale Face Database:**  The Yale Face database (Belhumeur et al., 1997) is widely used for computer vision and face recognition algorithms. There are 15 classes corresponding to 15 individuals' facee with 11 different facial expressions. Images are $320 \times 243$ in grayscale.

### 2.1  Data Preprocessing

No preprocessing is needed on MNIST, Fashion MNIST, or COIL-20. However, some preprocessing is needed for the Yale Face Database and 20-Newsgroups dataset, which we describe briefly here.

For the Yale Face dataset, MTCNN (Zhang et al., 2016) is used to crop the images. MTCNN crops the face of the image in three steps: In the first step, the model identifies the region that is likely contain faces. In the second step, the face image from the first step is refined, and if the selected region does not contain a face, then it gets eliminated. In the final step, facial landmarks (e.g. nose, eyes, lips) are detected, and the image is cropped to yield the final output. The cropped Yale Face dataset was resized to $64 \times 64$ pixels.

For the 20-Newsgroups dataset, we use tf-idf (Term Frequency-Inverse Document Frequency) vectorization to get a numeric data matrix for the document corpus, where the rows represent the document and the columns are unique words in the corpus. The matrix is tf-idf$(t, d) = $ tf$(t, d)$idf$(t)$, where tf$(t, d)$ is the number of appearences of term $t$ in document $d$, and idf$(t) = \log \frac{N}{\mathrm{df}(t)+1}$, where $N$ is the number of documents in the corpus, and df$(t)$ is the number of documents that contain the term $t$. Tokenization is the first step of vectorization, where sentences in documents are split into their meaningful units, typically words. In the next step, stop words are eliminated. Stop words are the most common words that are typically not significant for analysis. Examples of stop words in English include words like "the", "is", "and", "an", "on", etc. We also eliminate words with total frequency less than 5 across all documents, and then only the top

500 most frequent words of each document are used. After these steps, all unique words are collected and the frequency of all the words is counted for each document in the corpus, from which the tf-idf matrix is computed. Rows represent documents, and columns represent unique words in the corpus. We use the Natural Language ToolKit (NLTK) (Bird et al., 2009) for our implementation.

To calculate the Wasserstein distance matrix for newsgroup data set, we deployed the pytorch implementation of word2vec (Mikolov et al., 2013) to represent words in a vector space. One limitation of word2vec is that the vocabularies of the model are limited to the words that the model encountered during training. To address this, during the data preprocessing phase, we filter out words from the test data that do not appear in the training data. Consequently, some of the documents in the testing data set might become out of vocabulary (OOV) (Řehůřek & Sojka, 2010) words and eventually those documents were discarded from the test set.

### 2.2 Preprocessing for Wasserstein neighbor embeddings

Image data sets naturally lend themselves to Euclidean distance computations, as images are typically stored as vectors or matrices . One natural way to treat an image as a probability measure, thereby allowing one to compute Wasserstein distances between images, is to map the pixels to a discrete grid in $\mathbb{R}^2$ inside the unit cube $[0, 1]^2$. A dirac mass is then formed at the center of each grid square, and its mass is the corresponding pixel intensity. More concretely, a $28 \times 28$ image $A_{ij}$, $i, j = 1, \ldots, 28$ is mapped onto the grid with sides $\left[\frac{i}{28}, \frac{i+1}{28}\right] \times \left[\frac{j}{28}, \frac{j+1}{28}\right]$, $i, j = 0, \ldots, 27$ with centers $(x_i, y_j)$, and the measure formed is $\frac{1}{\sum_{i,j} A_{ij}} \sum_{i,j} A_{ij} \delta_{(x_i, y_j)}$. See (Hamm et al., 2022) for further details. In implementation, we store such a measure as a numpy array of size $28^2 \times 3$, with the first two entries being the centers $(x_i, y_j)$ and the last entry being the intensity $A_{ij}$.

To compute Wasserstein distances between two such measures, we use the python optimal transport (POT) package (Flamary et al., 2021). For most images, we use the linear program to compute the exact $W_2$ distance, but for larger scale images (in particular the Yale images here) we use the Sinkhorn algorithm based on entropic regularization to approximate the $W_2$ distance (Cuturi, 2013).

## 3 Supervised learning − classification

Classification remains a fundamental task in machine learning, including in computer vision and natural language processing (NLP), as it is key in many applications. The purpose of this section is to experimentally test when and why optimal transport-based dimensionality reduction outperforms Euclidean-based dimensionality reduction when performing classification. In our experiments, we use MDS, Isomap, t-SNE, and Laplacian eigenmaps (called spectral embedding in scikit-learn, where the underlying metric is either the Euclidean metric or the Wasserstein metric.

### 3.1 Methodology:

Our classification experiments run as follows: embed all of the data into lower-dimensional Euclidean space using either Euclidean or Wasserstein neighbor embeddings, split embedded data into a training and testing set, then classify using the train/test split.

For classification studies, we compare and hypothesize the accuracies using different datasets for optimal transport-based dimensionality reduction methods against Euclidean-based methods to check when the former one outperforms the latter and when it does not perform better. For the hypothesis test, we used a series of manifold learning techniques and classification algorithms. The classification algorithms we used are Linear Discriminant Analysis (LDA), k-Nearest Neighbors (KNN) with ($k = 1, 3, 5$), Support Vector Machine (SVM) with linear or RBF kernel, Random Forest (RF), and Multinomial Logistic Regression (MLR).

An important hyperparameter of dimensionality reduction methods is the embedding dimension itself. One way to choose this that we employ is to compute the SVD of the data matrix, then set the embedding

dimension $n$ to be the minimum integer such that

$$\frac{\sum_{i=1}^{n} \sigma_i^2}{\sum_{j=1}^{N} \sigma_j^2} \geq .95,$$

i.e., the number of singular values that capture at least 95% of the total spectrum (Medina et al., 2019).

For each pair of embedding and classification method, we compute the average accuracy over 10 replicate experiments. We then perform a 2-sample t-test to determine whether there is a significant improvement in accuracy of the Wasserstein embedding compared to Euclidean embedding. That is, we use

- **Null Hypothesis ($H_0$)**: The classification accuracy using the optimal transport-based dimensionality reduction method is less than or equal to the Euclidean-based methods.

- **Alternative Hypothesis ($H_a$):** The classification accuracy using the optimal transport-based dimensionality reduction method is better than the Euclidean-based dimensionality reduction methods.

We perform the hypothesis test with a significance level of $p = 0.05$.

The number of data points we used for Handwritten MNIST, and Fashion MNIST are 1000 and for Coil-20 is 1440. The proportion of the training and testing dataset for MNIST and Fashion MNIST is $70\% - 30\%$ (out of 1000 total points). For Coil-20, a total of 1000 points were used for training and 440 points were used to test the model. Of the total 1440 points, approximately 65% of the data is used for training and 35 to test the model.

### 3.2 Classification experimental results

Table 1 presents the classification accuracies for the MNIST data set for both types of embeddings. Rows correspond to different embedding algorithms (Spec Emb being Laplacian Eigenmaps), while columns are classification algorithms (abbreviations are LDA for linear discriminant analysis, KNN for k-nearest neighbors, SVD(Lin) and SVD(RBF) for linear and Gaussian kernel support vector machines, RF for random forest, and MLR for multinomial logistic regression). The accuracies of the optimal transport-based methods are on the right of the vertical line in each cell, and those of Euclidean-based methods are on the left. Entries are the mean of 10 independent trials of the classifier. Entries on the right of each cell are bolded if the result of the 2-sample t-test is to reject the null hypothesis, meaning that the optimal transport-based accuracy is statistically significantly better than the Euclidean-based accuracy for the given pair of embedding and classifier. An entry is not bolded if we fail to reject the null hypothesis. The results show that optimal transport-based embeddings generally achieve higher accuracy, though MDS and t-SNE embeddings exhibit more instances where the null hypothesis cannot be rejected.

Table 1: Comparison of Handwritten MNIST classification accuracies using optimal transport-based (Right) and Euclidean-based (Left) dimensionality reduction methods. Bold font indicates statistically significant improvement of optimal transport-based methods over Euclidean-based methods, as determined by a 2-sample t-test.

|  | LDA | 1NN | 3NN | 5NN | SVM(Lin) | SVM(RBF) | RF | MLR |
|---|---|---|---|---|---|---|---|---|
| **MDS** | 0.78 \| **0.86** | 0.83 \| 0.83 | 0.84 \| 0.83 | 0.83 \| 0.84 | 0.83 \| **0.90** | 0.88 \| **0.91** | 0.80 \| **0.86** | 0.76 \| **0.90** |
| **Isomap** | 0.84 \| **0.88** | 0.79 \| **0.84** | 0.80 \| **0.84** | 0.81 \| **0.85** | 0.84 \| **0.88** | 0.87 \| **0.90** | 0.83 \| **0.87** | 0.80 \| **0.89** |
| **t-SNE** | 0.74 \| 0.75 | 0.85 \| **0.87** | 0.85 \| **0.88** | 0.84 \| **0.88** | 0.77 \| 0.77 | 0.80 \| 0.80 | 0.85 \| **0.88** | 0.75 \| 0.75 |
| **Spec Emb** | 0.11 \| **0.85** | 0.11 \| **0.85** | 0.11 \| **0.84** | 0.11 \| **0.81** | 0.13 \| **0.87** | 0.11 \| **0.22** | 0.11 \| **0.89** | 0.11 \| **0.87** |

The performance of our method for the 20 Newsgroup datasets, shown in Table 2 is significantly better than the Euclidean-based methods for most cases (with some exception in the spectral embedding case). The trend continues for larger number of classes, including the full dataset, but are not shown for brevity. Experimental results for 5 classes are shown in the appendix (Table 14).

Table 2: Comparison of 20 Newsgroup classification accuracies using optimal transport-based (Right) and Euclidean-based (Left) dimensionality reduction methods using 2 classes (alt.atheism and sci.space) from 2 different broad categories. Bold font indicates statistically significant improvement of optimal transport-based methods over Euclidean-based methods, as determined by a 2-sample t-test.

| | LDA | 1NN | 3NN | 5NN | SVM(Lin) | SVM(RBF) | RF | MLR |
|---|---|---|---|---|---|---|---|---|
| **MDS** | 0.72 \| **0.83** | 0.67 \| **0.76** | 0.69 \| **0.80** | 0.70 \| **0.81** | 0.58 \| **0.82** | 0.74 \| **0.83** | 0.72 \| **0.80** | 0.70 \| **0.83** |
| **Isomap** | 0.75 \| **0.85** | 0.69 \| **0.81** | 0.71 \| **0.83** | 0.73 \| **0.84** | 0.72 \| **0.85** | 0.77 \| **0.86** | 0.74 \| **0.84** | 0.73 \| **0.86** |
| **t-SNE** | 0.73 \| **0.80** | 0.81 \| **0.86** | 0.78 \| **0.87** | 0.78 \| **0.87** | 0.75 \| **0.80** | 0.73 \| **0.87** | 0.73 \| **0.87** | 0.77 \| **0.80** |
| **Spec Emb** | 0.85 \| 0.85 | 0.75 \| **0.82** | 0.78 \| **0.85** | 0.79 \| **0.86** | 0.46 \| **0.86** | 0.84 \| 0.55 | 0.80 \| 0.55 | 0.48 \| **0.85** |

Table 3: Comparison of classification accuracies on the Fashion MNIST dataset using optimal transport-based (Right) and Euclidean-based (Left) dimensionality reduction techniques. Bold text highlights cases where optimal transport-based methods significantly outperform Euclidean-based methods, as confirmed by a two-sample t-test.

| | LDA | 1NN | 3NN | 5NN | SVM(Lin) | SVM(RBF) | RF | MLR |
|---|---|---|---|---|---|---|---|---|
| **MDS** | 0.72 \| 0.68 | 0.69 \| 0.68 | 0.69 \| 0.68 | 0.70 \| 0.68 | 0.71 \| **0.72** | 0.74 \| 0.71 | 0.72 \| 0.70 | 0.65 \| **0.71** |
| **Isomap** | 0.74 \| 0.67 | 0.70 \| 0.65 | 0.71 \| 0.68 | 0.72 \| 0.68 | 0.69 \| 0.69 | 0.74 \| 0.69 | 0.73 \| 0.69 | 0.68 \| **0.69** |
| **t-SNE** | 0.62 \| 0.61 | 0.73 \| 0.65 | 0.72 \| 0.66 | 0.71 \| 0.66 | 0.65 \| 0.64 | 0.67 \| 0.65 | 0.72 \| 0.67 | 0.63 \| 0.61 |
| **Spec Emb** | 0.09 \| **0.15** | 0.10 \| **0.68** | 0.10 \| **0.66** | 0.09 \| **0.65** | 0.09 \| **0.70** | 0.08 \| 0.09 | 0.09 \| **0.72** | 0.07 \| **0.17** |

In contrast, classification results for Fashion MNIST (Table 3) and Coil-20 (Table 4) are not as good using our method. In particular, for most embeddings and classifiers, we do not see significant improvement using optimal transport.

From these experimental results, it appears that the structure of the data set itself can have significant impact on which distance metric (Wasserstein or Euclidean) provides a more meaningful notion of geometry. This is not necessarily surprising, but it is worth considering what facets of the data contribute to success when using one metric or another. The Coil-20 data set consists of images of objects rotated from 0° to 360° (Figure 1). Optimal transport treats each image as a probability measure (with mass 1), and finds the optimal deformation to go from one image to the other. However, looking at the elements from the same class in Figure 1, we see that rotated copies of the bottle could well have different masses (hence be finite measures but not probabilities). Alternatively, rotated copies of the bottle may not be easily morphed into each other using optimal transport. Consequently, it may be that standard balanced optimal transport distances do not give a precise enough description of the geometry of Coil-20. We speculate that unbalanced optimal transport (Chizat et al., 2018), in which one can create and destroy mass and align measures that are not probabilities, could give a better notion of geometry for a rotational data set like Coil-20.

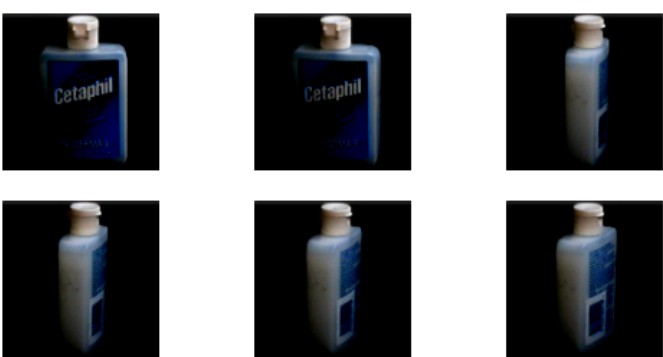

Figure 1: Six images that are from the same class of Coil-20 data set that comprised with a rotation of 5° at a time for 0° - 360° that produces 72 images for a class.

Table 4: Comparison of classification accuracies on the Coil-20 dataset using optimal transport-based (Right) and Euclidean-based (Left) dimensionality reduction techniques. Bold text highlights cases where optimal transport-based methods significantly outperform Euclidean-based methods, as confirmed by a two-sample t-test.

|  | LDA | 1NN | 3NN | 5NN | SVM(Lin) | SVM(RBF) | RF | MLR |
|---|---|---|---|---|---|---|---|---|
| MDS | 0.42 \| **0.53** | 0.76 \| 0.75 | 0.77 \| 0.76 | 0.76 \| 0.75 | 0.07 \| **0.10** | 0.65 \| **0.67** | 0.76 \| 0.76 | 0.17 \| **0.17** |
| Isomap | 0.59 \| 0.51 | 0.74 \| 0.72 | 0.72 \| 0.68 | 0.71 \| 0.67 | 0.41 \| 0.31 | 0.62 \| 0.53 | 0.74 \| 0.72 | 0.41 \| **0.30** |
| t-SNE | 0.82 \| 0.59 | 0.99 \| 0.87 | 0.97 \| 0.84 | 0.96 \| 0.82 | 0.84 \| 0.65 | 0.86 \| 0.68 | 0.98 \| 0.84 | 0.83 \| 0.61 |
| Spec Emb | 0.04 \| **0.53** | 0.06 \| **0.72** | 0.05 \| **0.72** | 0.05 \| **0.72** | 0.03 \| **0.04** | 0.04 \| **0.64** | 0.06 \| **0.73** | 0.03 \| **0.04** |

Fashion MNIST has a couple of features that we speculate lead to a lack of increase in performance for optimal transport-based embeddings: 1) variation within classes, and 2) significantly different texture signals in the data. Figure 2 shows a few images from the "dress" class, in which we see both the varying textures as well as varying morphology of elements within the same class.

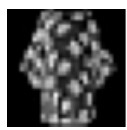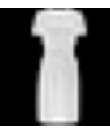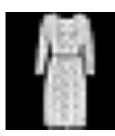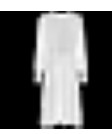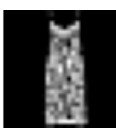

Figure 2: Images of Fashion MNIST dataset for the class label Dress.

# 4 Unsupervised learning − clustering

## 4.1 Clustering Algorithms

Our second set of experiments is for data clustering. We utilize $k$-means clustering and spectral clustering based on the eigendecomposition of the graph Laplacian $L = D - W$, where $D$ is the degree matrix, and $W$ is an adjacency matrix for a graph supported on the data.

## 4.2 Methodology

We compare clustering accuracy between the optimal transport-based embeddings and Euclidean-based embeddings. While clustering algorithms do not use labels, we keep the labels to evaluate the accuracy of our clustering procedure. The clustering process follows three steps: 1) compute the distance matrix $W$ using the Wasserstein metric, 2) embed the data using one of the dimensionality reduction algorithms, and (3) cluster the embedded data. We use linear sum assignment (Kuhn, 1955) to map the cluster labels onto the ground truth labels in the optimal way.

## 4.3 Experimental Results

We conducted clustering experiments on 20-Newsgroups, MNIST, Fashion MNIST, and Yale Face datasets. For 20-Newsgroups, we ran 3 experiments on 2, 5, and all 20 classes, sampling 100 documents per class for the full dataset due to runtime constraints. After preprocessing, document counts were slightly reduced due to out-of-vocabulary words. MNIST and Fashion MNIST experiments used 1,000 data points each, while Yale Face experiments used 3, 8, and all 15 classes.

Tables 5, 6, and 7 summarize these results. The optimal transport-based method generally outperformed Euclidean embeddings, particularly for MNIST and Fashion MNIST with spectral clustering. However, for some cases, we failed to reject the null hypothesis. We see improvement for many cases in the 20-Newsgroups clustering; however, we note that overall clustering performance for the full dataset is low, so further pre-processing would be needed to make accuracy competitive. Yale shows the least cases of improvement using

optimal transport. This is potentially due to the cropping preprocessing step and variations among textures and face morphology.

Table 5: Clustering results for MNIST and Fashion MNIST. The 2nd and 3rd columns represent the clustering accuracies using Euclidean-based (left) and optimal transport-based (right) methods separated by vertical lines. Bold text highlights cases where optimal transport-based methods significantly outperform Euclidean-based methods, as confirmed by a two-sample t-test.

| Method | Clustering Algorithm | Accuracy | |
|---|---|---|---|
| | | MNIST | Fashion MNIST |
| Isomap | $k$-means | 0.55 \| **0.59** | 0.53 \| 0.54 |
| | Spectral | 0.14 \| **0.67** | 0.13 \| **0.51** |
| MDS | $k$-means | 0.55 \| 0.40 | 0.52 \| 0.45 |
| | Spectral | 0.13 \| **0.54** | 0.13 \| **0.35** |
| t-SNE | $k$-means | 0.65 \| 0.67 | 0.59 \| 0.59 |
| | Spectral | 0.55 \| **0.65** | 0.53 \| **0.57** |

Table 6: Clustering results for 20 Newsgroups dataset using 2 classes (alt. atheism and sci. space), 5 classes (alt. atheism, comp.sys., mac.hardware, rec.sport.baseball, sci.space, and talk.politics.guns) from each of the 5 different broader categories (alternative, computer, mac, recreation, science, and talk), and all 20 classes. The clustering accuracies are separated by a vertical line for Euclidean-based (left) and optimal transport-based (right) methods. Bold text highlights cases where optimal transport-based methods significantly outperform Euclidean-based methods, as confirmed by a two-sample t-test.

| Method | Clustering Algorithm | Accuracy | | |
|---|---|---|---|---|
| | | 2 classes | 5 classes | all classes |
| Isomap | $k$-means | 0.55 \| **0.59** | 0.53 \| 0.48 | 0.16 \| **0.18** |
| | Spectral | 0.59 \| 0.57 | 0.53 \| 0.49 | 0.17 \| **0.18** |
| MDS | $k$-means | 0.55 \| **0.61** | 0.24 \| **0.29** | 0.10 \| **0.12** |
| | Spectral | 0.54 \| **0.69** | 0.24 \| **0.40** | 0.10 \| **0.14** |
| t-SNE | $k$-means | 0.61 \| 0.59 | 0.60 \| 0.46 | 0.26 \| 0.17 |
| | Spectral | 0.59 \| 0.57 | 0.55 \| 0.50 | 0.26 \| 0.18 |

Table 7: Clustering results for Yale Face dataset using 3 classes, 8 classes, and all 15 classes. A vertical line is used to separate the accuracies for Euclidean-based (left) and optimal transport-based (right) methods. Bold text highlights cases where optimal transport-based methods significantly outperform Euclidean-based methods, as confirmed by a two-sample t-test.

| Method | Clustering Algorithm | Accuracy | | |
|---|---|---|---|---|
| | | 3 classes | 8 classes | all 15 classes |
| Isomap | $k$-means | 0.70 \| 0.67 | 0.62 \| 0.59 | 0.49 \| 0.46 |
| | Spectral | 0.88 \| 0.88 | 0.58 \| **0.72** | 0.57 \| **0.60** |
| MDS | $k$-means | 0.72 \| 0.43 | 0.63 \| 0.48 | 0.56 \| 0.27 |
| | Spectral | 0.81 \| **0.85** | 0.69 \| 0.59 | 0.64 \| 0.56 |
| t-SNE | $k$-means | 0.60 \| 0.46 | 0.63 \| 0.60 | 0.61 \| 0.59 |
| | Spectral | 0.62 \| 0.49 | 0.65 \| 0.62 | 0.66 \| 0.64 |

## 5 Computational Considerations

The main challenge of computing Wasserstein neighbor embeddings is that a single Wasserstein distance computation between $n$-pixel images costs $\Omega(n^3 \log n)$ flops to compute exactly for dimensions $d > 1$ Peyré & Cuturi (2019). Entropic regularization can improve this (Cuturi, 2013), and therefore we use the sinkhorn

distance to approximate all $W_2$ distances computed above. However, largescale or high-resolution datasets pose additional challenges. For the latter, one can downsample the images to lower resolution. In our experience, unless one downsamples dramatically, this does not significantly adversely affect the quality of the Wasserstein embedding. For datasets with many points, one needs to either subsample the data itself, or reduce the number of distance computations.

For our experiments above, we randomly subsample large data sets to perform the embedding and classification or clustering on a subset of the data (see Section 3 for details). For the Yale face dataset, images are natively $320 \times 243$ but are cropped and downsampled to a resolution of $64 \times 64$ (see Section 2.1). For Coil-20, we utilized another approximation step in the Wasserstein embedding. Namely, we used the Nyström method (Williams & Seeger, 2000) to approximate the Wasserstein distance matrix. We randomly compute 80 columns out of 1440, and form the standard Nyström approximation (without any further rank truncation). This matrix is then used to compute the embedding. This reduces the total number of Sinkhorn computations from approximately $1,000,000$ to approximately $100,000$; a reduction of an order of magnitude.

We note that the word-mover's distance used for the Newsgroup embeddings only requires computing 1-dimensional Wasserstein distances, which are fast to compute. Therefore, we compute the full Wasserstein distances between all documents used in the experiments.

## 6 Conclusion

We have tested the hypothesis that modeling data in a dimensionality reduction pipeline using the geometry of Wasserstein space leads to better clustering and classification performance. Overall, we see that this is true for some datasets and not for others. In particular, we see success of optimal transport-based methods on the simple, curated dataset MNIST and the NLP dataset 20 Newsgroup both for clustering and classification. However, for Fashion MNIST, Coil-20, and Yale face dataset, we see mixed performance of the method, potentially due to textured signals, rotational variation, and varying morphology among classes. It seems that NLP data is a better candidate in general for use of Wasserstein distance, as noted in work on the word mover's distance Kusner et al. (2015) and the experiments here. For images, it appears that varying textures and significant backgrounds can cause issues. Smoothing textures could be one approach to consider in the future to mitigate that issue, whereas foreground/background segmentation could be used on images with background variations. Finally, future work will consider unbalanced optimal transport, which can better account for some transformations of images, to determine if it yields better performance on a wider variety of data.

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

# A    Appendix

## A.1    Classification experimental results

Table 8 and 9 represents the classification results for OT-based and Euclidean-based techniques for the MNIST data set. Table 10 shows the results of the 2-sample t-test of MNIST data for OT-based method and Euclidean-based method. See Table 1 and Sections 3.2 and 3.1 of the main paper for further details on MNIST classification results and experiment setup.

Table 8: Classification accuracy for Handwritten dataset utilizing optimal transport-based embedding.

|  | LDA | 1NN | 3NN | 5NN | SVM(Lin) | SVM(RBF) | RF | MLR |
|---|---|---|---|---|---|---|---|---|
| **MDS** | $0.86 \pm 0.03$ | $0.83 \pm 0.03$ | $0.83 \pm 0.02$ | $0.84 \pm 0.03$ | $0.90 \pm 0.02$ | $0.91 \pm 0.02$ | $0.86 \pm 0.03$ | $0.90 \pm 0.01$ |
| **Isomap** | $0.88 \pm 0.03$ | $0.84 \pm 0.02$ | $0.84 \pm 0.02$ | $0.85 \pm 0.02$ | $0.88 \pm 0.02$ | $0.90 \pm 0.01$ | $0.87 \pm 0.01$ | $0.89 \pm 0.02$ |
| **t-SNE** | $0.75 \pm 0.02$ | $0.87 \pm 0.02$ | $0.88 \pm 0.02$ | $0.88 \pm 0.02$ | $0.77 \pm 0.03$ | $0.80 \pm 0.03$ | $0.88 \pm 0.02$ | $0.75 \pm 0.03$ |
| **Spec Emb** | $0.85 \pm 0.01$ | $0.85 \pm 0.03$ | $0.84 \pm 0.01$ | $0.81 \pm 0.02$ | $0.87 \pm 0.02$ | $0.22 \pm 0.04$ | $0.89 \pm 0.02$ | $0.87 \pm 0.02$ |

Table 9: Classification accuracy for Handwritten MNIST dataset using Euclidean based embedding.

|  | LDA | 1NN | 3NN | 5NN | SVM(Lin) | SVM(RBF) | RF | MLR |
|---|---|---|---|---|---|---|---|---|
| **MDS** | $0.78 \pm 0.04$ | $0.83 \pm 0.02$ | $0.84 \pm 0.02$ | $0.83 \pm 0.02$ | $0.83 \pm 0.03$ | $0.88 \pm 0.03$ | $0.80 \pm 0.03$ | $0.76 \pm 0.04$ |
| **Isomap** | $0.84 \pm 0.03$ | $0.79 \pm 0.02$ | $0.80 \pm 0.03$ | $0.81 \pm 0.03$ | $0.84 \pm 0.03$ | $0.87 \pm 0.03$ | $0.83 \pm 0.03$ | $0.80 \pm 0.02$ |
| **t-SNE** | $0.74 \pm 0.04$ | $0.85 \pm 0.01$ | $0.85 \pm 0.02$ | $0.84 \pm 0.02$ | $0.77 \pm 0.03$ | $0.80 \pm 0.03$ | $0.85 \pm 0.02$ | $0.75 \pm 0.03$ |
| **Spec Emb** | $0.11 \pm 0.01$ | $0.11 \pm 0.01$ | $0.11 \pm 0.02$ | $0.11 \pm 0.02$ | $0.13 \pm 0.01$ | $0.11 \pm 0.01$ | $0.10 \pm 0.02$ | $0.11 \pm 0.01$ |

Table 10: Two-sample t-test comparison between optimal transport and Euclidean-based methods for the MNIST dataset. Shown are the $t$ statistic and the result of the hypothesis test.

| Method | MDS | Isomap | t-SNE | Spec Emb |
|---|---|---|---|---|
| LDA | 7.16, 0.00, Reject $H_0$ | 4.22, 0.00, Reject $H_0$ | 1.00, 0.16, Fail to Reject $H_0$ | 234.01, 0.00, Reject $H_0$ |
| 1NN | 0.00, 0.50, Fail to Reject $H_0$ | 7.91, 0.00, Reject $H_0$ | 4.00, 0.00, Reject $H_0$ | 104.65, 0.00, Reject $H_0$ |
| 3NN | -1.58, 0.94, Fail to Reject $H_0$ | 4.96, 0.00, Reject $H_0$ | 4.74, 0.00, Reject $H_0$ | 146.00, 0.00, Reject $H_0$ |
| 5NN | 1.24, 0.11, Fail to Reject $H_0$ | 4.96, 0.00, Reject $H_0$ | 6.32, 0.00, Reject $H_0$ | 140.00, 0.00, Reject $H_0$ |
| SVM(Lin) | 8.68, 0.00, Reject $H_0$ | 4.96, 0.00, Reject $H_0$ | 0.00, 0.50, Fail to Reject $H_0$ | 148.00, 0.00, Reject $H_0$ |
| SVM(RBF) | 3.72, 0.00, Reject $H_0$ | 4.24, 0.00, Reject $H_0$ | 0.50, 0.50, Fail to Reject $H_0$ | 11.93, 0.00, Reject $H_0$ |
| RF | 6.32, 0.00, Reject $H_0$ | 5.66, 0.00, Reject $H_0$ | 4.74, 0.00, Reject $H_0$ | 124.91, 0.00, Reject $H_0$ |
| MLR | 15.19, 0.00, Reject $H_0$ | 14.23, 0.00, Reject $H_0$ | 0.50, 0.50, Fail to Reject $H_0$ | 152.00, 0.00, Reject $H_0$ |

Tables 11 and 12 show the classification results for the newsgroup data set using 2 classes for the OT-based method and Euclidean-based method respectively. Table 13 represents the results of the 2-sample hypothesis of the newsgroup data set using 2 classes. We also carried out a classification experiment using 5 classes of newsgroup data set. Tables 15, 16 and 17 show the OT-based, Euclidean-based classification results and 2-sample t-test results for 5 classes of the newsgroup data set respectively. See Table 2 for the classification results of the newsgroup data set using 2 classes for the OT-based and Euclidean-based method. The classification results for the Fashion MNIST data set are represented in Tables 18, 19 for OT-based and Euclidean-based respectively. Table 20 shows the results of the 2-sample t-test for the Fashion MNIST data set. See Table 3 for the comparison results of the OT-based and Euclidean-based method of the Fashion MNIST data set. The classification results for the coil-20 data set are represented in Table 21, 22 for the OT-based and Euclidean-based method. Table 23 represents the results of 2-sample t-test results for the coil-20 data set. See Table 4 for the OT-based and Euclidean-based compared results of the coil-20 data set and the discussion of the results.

Table 11: Classification accuracy for 20 Newsgroups dataset using optimal transport-based embedding using 2 classes (alt. atheism and sci. space).

|  | LDA | 1NN | 3NN | 5NN | SVM(Lin) | SVM(RBF) | RF | MLR |
|---|---|---|---|---|---|---|---|---|
| **MDS** | 0.83 ± 0.05 | 0.76 ± 0.03 | 0.80 ± 0.03 | 0.81 ± 0.02 | 0.82 ± 0.05 | 0.83 ± 0.03 | 0.80 ± 0.04 | 0.83 ± 0.04 |
| **Isomap** | 0.85 ± 0.02 | 0.81 ± 0.03 | 0.83 ± 0.03 | 0.84 ± 0.03 | 0.85 ± 0.02 | 0.86 ± 0.03 | 0.84 ± 0.03 | 0.86 ± 0.02 |
| **t-SNE** | 0.80 ± 0.05 | 0.86 ± 0.03 | 0.87 ± 0.04 | 0.87 ± 0.03 | 0.80 ± 0.05 | 0.87 ± 0.04 | 0.87 ± 0.05 | 0.80 ± 0.05 |
| **Spec Emb** | 0.85 ± 0.01 | 0.82 ± 0.02 | 0.85 ± 0.01 | 0.86 ± 0.01 | 0.86 ± 0.01 | 0.55 ± 0.01 | 0.55 ± 0.01 | 0.85 ± 0.01 |

Table 12: Classification accuracy for 20 Newsgroups dataset using Euclidean based embedding using 2 classes (alt. atheism, sci. space).

|  | LDA | 1NN | 3NN | 5NN | SVM(Lin) | SVM(RBF) | RF | MLR |
|---|---|---|---|---|---|---|---|---|
| **MDS** | 0.72 ± 0.04 | 0.67 ± 0.04 | 0.69 ± 0.04 | 0.70 ± 0.03 | 0.58 ± 0.09 | 0.74 ± 0.04 | 0.72 ± 0.03 | 0.70 ± 0.04 |
| **Isomap** | 0.75 ± 0.01 | 0.69 ± 0.01 | 0.71 ± 0.01 | 0.73 ± 0.01 | 0.72 ± 0.01 | 0.77 ± 0.01 | 0.74 ± 0.01 | 0.73 ± 0.01 |
| **t-SNE** | 0.73 ± 0.02 | 0.81 ± 0.01 | 0.78 ± 0.01 | 0.78 ± 0.01 | 0.75 ± 0.01 | 0.73 ± 0.02 | 0.73 ± 0.03 | 0.77 ± 0.01 |
| **Spec Emb** | 0.85 ± 0.02 | 0.75 ± 0.02 | 0.78 ± 0.02 | 0.79 ± 0.02 | 0.46 ± 0.03 | 0.84 ± 0.02 | 0.80 ± 0.02 | 0.48 ± 0.09 |

## A.2 Clustering results

Table 24 represents the clustering results for MNIST and Fashion MNIST. In these tables, we show the average clustering accuracies with the standard deviation of 10 trials. Tables 25 and 26 show the results of 2-sample t-test for clustering for the MNIST adn Fashion MNIST data set respectively. See Table 5 for the comparison of the results for the MNIST and Fashion MNIST data set and Section 4.2 and 4.3 for further details. Table 27 shows the clustering accuracies of the 20 Newsgroup data set for $2, 5$ and all 20 classes and Tables 28, 29 and 30 represent the 2 sample hypothesis test results of the corresponding data set. See Table 6 in the main paper for the clustering comparison of the 20 Newsgroup data set. Table 31 shows the clustering accuracies of Yale Face data using 3, 8 and all 15 classes of the data set. In Tables 32, 33 and 34 we represent the results of the 2 sample hypothesis test using 3, 5 and all 15 classes of the Yale Face data set respectively. See Table 7 for comparison results of the OT-based method and the Euclidean-based method for cluster accuracies of Yale Face data set.

Table 13: Two-sample t-test comparison between optimal transport and Euclidean-based methods for the 20 Newsgroup dataset using 2 classes (alt. atheism and sci. space) from 2 broader categories (alternatives and science). Shown are the $t$ statistic and the result of the hypothesis test.

| Methods | MDS | Isomap | t-SNE | Spec Emb |
|---|---|---|---|---|
| LDA | 6.67, 0.00, Reject $H_0$ | 16.07, 0.00, Reject $H_0$ | 3.54, 0.00, Reject $H_0$ | 0.12, 0.45, Fail to Reject $H_0$ |
| 1NN | 5.88, 0.00, Reject $H_0$ | 10.67, 0.00, Reject $H_0$ | 3.78, 0.00, Reject $H_0$ | 8.56, 0.00, Reject $H_0$ |
| 3NN | 6.75, 0.00, Reject $H_0$ | 11.39, 0.00, Reject $H_0$ | 5.79, 0.00, Reject $H_0$ | 8.46, 0.00, Reject $H_0$ |
| 5NN | 8.53, 0.00, Reject $H_0$ | 12.68, 0.00, Reject $H_0$ | 10.01, 0.00, Reject $H_0$ | 9.47, 0.00, Reject $H_0$ |
| SVM(Lin) | 4.82, 0.00, Reject $H_0$ | 10.06, 0.00, Reject $H_0$ | 2.80, 0.00, Reject $H_0$ | 40.49, 0.00, Reject $H_0$ |
| SVM(RBF) | 8.21, 0.00, Reject $H_0$ | 13.84, 0.00, Reject $H_0$ | 10.43, 0.00, Reject $H_0$ | -38.03, 1.00, Fail to Reject $H_0$ |
| RF | 5.77, 0.00, Reject $H_0$ | 9.72, 0.00, Reject $H_0$ | 8.10, 0.00, Reject $H_0$ | -36.24, 1.00, Fail to Reject $H_0$ |
| MLR | 6.59, 0.00, Reject $H_0$ | 14.82, 0.00, Reject $H_0$ | 1.88, 0.00, Reject $H_0$ | 13.29, 0.00, Reject $H_0$ |

Table 14: Comparison of 20 Newsgroup classification accuracies using optimal transport-based (Right) and Euclidean-based (Left) dimensionality reduction methods using 5 random classes from 5 different broader categories. Bold font indicates statistically significant improvement of optimal transport-based methods over Euclidean-based methods, as determined by a 2-sample t-test.

| | LDA | 1NN | 3NN | 5NN | SVM Linear | SVM RBF | RF | MLR |
|---|---|---|---|---|---|---|---|---|
| MDS | 0.36 \| **0.43** | 0.30 \| **0.34** | 0.32 \| **0.37** | 0.34 \| **0.38** | 0.24 \| **0.43** | 0.24 \| **0.43** | 0.38 \| **0.43** | 0.35 \| **0.43** |
| Isomap | 0.40 \| **0.64** | 0.35 \| **0.55** | 0.37 \| **0.59** | 0.39 \| **0.60** | 0.37 \| **0.65** | 0.43 \| **0.66** | 0.40 \| **0.64** | 0.39 \| **0.64** |
| t-SNE | 0.35 \| **0.46** | 0.63 \| **0.67** | 0.55 \| **0.69** | 0.53 \| **0.69** | 0.36 \| **0.49** | 0.43 \| **0.61** | 0.55 \| **0.69** | 0.35 \| **0.46** |
| Spec Emb | 0.47 \| **0.55** | 0.43 \| **0.54** | 0.42 \| **0.55** | 0.45 \| **0.57** | 0.18 \| **0.20** | 0.52 \| **0.60** | 0.49 \| **0.60** | 0.19 \| 0.21 |

Table 15: Classification accuracy for 20 Newsgroups dataset using optimal transport-based embedding using 5 classes 5 classes (alt. atheism, comp.sys., mac.hardware, rec.sport.baseball, sci.space, and talk.politics.guns) from 5 different broader categories (alternative, computer, mac, recreation, science and talk).

| | LDA | 1NN | 3NN | 5NN | SVM(Lin) | SVM(RBF) | RF | MLR |
|---|---|---|---|---|---|---|---|---|
| MDS | 0.43 ± 0.04 | 0.34 ± 0.03 | 0.37 ± 0.03 | 0.38 ± 0.04 | 0.43 ± 0.03 | 0.43 ± 0.03 | 0.43 ± 0.02 | 0.43 ± 0.04 |
| Isomap | 0.64 ± 0.04 | 0.55 ± 0.03 | 0.59 ± 0.03 | 0.60 ± 0.03 | 0.65 ± 0.03 | 0.66 ± 0.03 | 0.64 ± 0.03 | 0.64 ± 0.03 |
| t-SNE | 0.46 ± 0.06 | 0.67 ± 0.03 | 0.69 ± 0.03 | 0.69 ± 0.03 | 0.49 ± 0.07 | 0.61 ± 0.03 | 0.69 ± 0.02 | 0.46 ± 0.06 |
| Spec Emb | 0.55 ± 0.01 | 0.54 ± 0.01 | 0.55 ± 0.01 | 0.57 ± 0.01 | 0.20 ± 0.01 | 0.60 ± 0.01 | 0.60 ± 0.01 | 0.21 ± 0.01 |

Table 16: Classification accuracy for 20 Newsgroups dataset using Euclidean-based embedding using 5 classes 5 classes (alt. atheism, comp.sys., mac.hardware, rec.sport.baseball, sci.space, and talk.politics.guns) from 5 different broader categories (alternative, computer, mac, recreation, science and talk).

| | LDA | 1NN | 3NN | 5NN | SVM(Lin) | SVM(RBF) | RF | MLR |
|---|---|---|---|---|---|---|---|---|
| MDS | 0.36 ± 0.04 | 0.30 ± 0.03 | 0.32 ± 0.02 | 0.34 ± 0.02 | 0.24 ± 0.05 | 0.24 ± 0.05 | 0.38 ± 0.03 | 0.35 ± 0.03 |
| Isomap | 0.40 ± 0.01 | 0.35 ± 0.01 | 0.37 ± 0.01 | 0.39 ± 0.01 | 0.37 ± 0.01 | 0.43 ± 0.01 | 0.40 ± 0.01 | 0.39 ± 0.01 |
| t-SNE | 0.35 ± 0.01 | 0.63 ± 0.01 | 0.55 ± 0.01 | 0.53 ± 0.01 | 0.36 ± 0.01 | 0.43 ± 0.01 | 0.55 ± 0.01 | 0.35 ± 0.01 |
| Spec Emb | 0.47 ± 0.03 | 0.43 ± 0.02 | 0.42 ± 0.02 | 0.45 ± 0.02 | 0.18 ± 0.02 | 0.52 ± 0.03 | 0.49 ± 0.03 | 0.19 ± 0.04 |

Table 17: Two-sample t-test comparison between optimal transport and Euclidean-based methods for the 20 Newsgroup dataset using 5 classes 5 classes (alt. atheism, comp.sys., mac.hardware, rec.sport.baseball, sci.space, and talk.politics.guns) from 5 different broader categories (alternative, computer, mac, recreation, science and talk). Shown are the $t$ statistic and the result of the hypothesis test.

| Methods | MDS | Isomap | t-SNE | Spec Emb |
|---|---|---|---|---|
| LDA | 4.03, 0.00, Reject $H_0$ | 20.03, 0.00, Reject $H_0$ | 5.17, 0.00, Reject $H_0$ | 8.06, 0.0, Reject $H_0$ |
| 1NN | 2.86, 0.00, Reject $H_0$ | 24.37, 0.00, Reject $H_0$ | 6.55, 0.00, Reject $H_0$ | 14.08, 0.00, Reject $H_0$ |
| 1NN | 4.34, 0.00, Reject $H_0$ | 21.99, 0.00, Reject $H_0$ | 15.49, 0.00, Reject $H_0$ | 20.07, 0.00, Reject $H_0$ |
| 5NN | 2.38, 0.01, Reject $H_0$ | 24.73, 0.00, Reject $H_0$ | 19.44, 0.00, Reject $H_0$ | 19.09, 0.00, Reject $H_0$ |
| SVM(Lin) | 9.49, 0.00, Reject $H_0$ | 27.80, 0.00, Reject $H_0$ | 6.36, 0.00, Reject $H_0$ | 2.5, 0.01, Reject $H_0$ |
| SVM(RBF) | 10.08, 0.00, Reject $H_0$ | 22.68, 0.00, Reject $H_0$ | 16.47, 0.00, Reject $H_0$ | 8.16, 0.00, Reject $H_0$ |
| RF | 4.71, 0.00, Reject $H_0$ | 24.60, 0.00, Reject $H_0$ | 23.35, 0.00, Reject $H_0$ | 12.53, 0.00, Reject $H_0$ |
| MLR | 5.08, 0.00, Reject $H_0$ | 24.00, 0.00, Reject $H_0$ | 5.13, 0.00, Reject $H_0$ | 1.00, 0.16, Fail to Reject $H_0$ |

Table 18: Classification accuracies for optimal transport-based embedding on Fashion MNIST dataset.

| | LDA | 1NN | 3NN | 5NN | SVM(Lin) | SVM(RBF) | RF | MLR |
|---|---|---|---|---|---|---|---|---|
| MDS | $0.68 \pm 0.02$ | $0.68 \pm 0.02$ | $0.68 \pm 0.02$ | $0.68 \pm 0.02$ | $0.72 \pm 0.02$ | $0.71 \pm 0.01$ | $0.70 \pm 0.02$ | $0.71 \pm 0.01$ |
| LSOMAP | $0.67 \pm 0.02$ | $0.65 \pm 0.02$ | $0.68 \pm 0.02$ | $0.68 \pm 0.02$ | $0.69 \pm 0.03$ | $0.69 \pm 0.01$ | $0.69 \pm 0.02$ | $0.69 \pm 0.02$ |
| t-SNE | $0.61 \pm 0.01$ | $0.65 \pm 0.02$ | $0.66 \pm 0.01$ | $0.66 \pm 0.02$ | $0.64 \pm 0.02$ | $0.65 \pm 0.02$ | $0.67 \pm 0.01$ | $0.61 \pm 0.02$ |
| Spec Emb | $0.15 \pm 0.06$ | $0.68 \pm 0.01$ | $0.66 \pm 0.01$ | $0.65 \pm 0.02$ | $0.70 \pm 0.03$ | $0.09 \pm 0.02$ | $0.72 \pm 0.01$ | $0.17 \pm 0.06$ |

Table 19: Euclidean-based embeddings classification accuracy for Fashion MNIST dataset.

| | LDA | 1NN | 3NN | 5NN | SVM(Lin) | SVM(RBF) | RF | MLR |
|---|---|---|---|---|---|---|---|---|
| MDS | $0.72 \pm 0.01$ | $0.69 \pm 0.01$ | $0.69 \pm 0.02$ | $0.70 \pm 0.02$ | $0.71 \pm 0.01$ | $0.74 \pm 0.03$ | $0.72 \pm 0.02$ | $0.65 \pm 0.02$ |
| LSOMAP | $0.74 \pm 0.01$ | $0.70 \pm 0.01$ | $0.71 \pm 0.02$ | $0.72 \pm 0.02$ | $0.69 \pm 0.02$ | $0.74 \pm 0.02$ | $0.73 \pm 0.02$ | $0.68 \pm 0.01$ |
| t-SNE | $0.62 \pm 0.02$ | $0.73 \pm 0.02$ | $0.72 \pm 0.01$ | $0.71 \pm 0.02$ | $0.65 \pm 0.02$ | $0.67 \pm 0.01$ | $0.72 \pm 0.01$ | $0.63 \pm 0.02$ |
| Spec Emb | $0.09 \pm 0.02$ | $0.10 \pm 0.02$ | $0.10 \pm 0.00$ | $0.09 \pm 0.01$ | $0.09 \pm 0.02$ | $0.08 \pm 0.02$ | $0.09 \pm 0.01$ | $0.07 \pm 0.01$ |

Table 20: Two-sample t-test comparison between optimal transport and Euclidean-based methods for the Fashion MNIST dataset

| Method | MDS | Isomap | t-SNE | Spec Emb |
|---|---|---|---|---|
| LDA | -8.00, 1.00, Fail to Reject $H_0$ | -14.0,1.00, Fail to Reject H0 | -2.00, 0.97, Fail to Reject $H_0$ | 4.24, 0.00, Reject $H_0$ |
| 1NN | -2.00, 0.97, Fail to Reject $H_0$ | -10.00, 1.00, Fail to Reject $H_0$ | -12.65, 1.00, Fail to Reject $H_0$ | 116.00, 0.00, Reject $H_0$ |
| 3NN | -1.58, 0.94, Fail to Reject $H_0$ | -4.74,1.00, Fail to Reject $H_0$ | -18.97, 1.00, Fail to Reject $H_0$ | 250.44, 0.00, Reject $H_0$ |
| 5NN | -3.16, 1.00, Fail to Reject $H_0$ | -6.32, 1.00, Fail to Reject $H_0$ | -7.91, 1.00, Fail to Reject $H_0$ | 112.00, 0.00, Reject $H_0$ |
| SVM(Lin) | 2.00, 0.03, Reject $H_0$ | 0.00, 0.50, Fail to Reject $H_0$ | -1.58, 0.94, Fail to Reject $H_0$ | 75.66, 0.00, Reject $H_0$ |
| SVM(RBF) | -4.24,1.00, Fail to Reject $H_0$ | -10.00, 1.00, Fail to Reject H0 | -4.00, 1.00, Fail to Reject $H_0$ | 1.58, 0.06, Fail to Reject $H_0$ |
| RF | -3.16,1.00, Fail to Reject $H_0$ | -6.32, 1.00, Fail to Reject $H_0$ | -15.81, 1.00, Fail to Reject $H_0$ | 199.22, 0.00, Reject $H_0$ |
| MLR | 12.00, 0.00, Reject $H_0$ | 2.00, 0.03, Reject $H_0$ | -3.16,1.00, Fail to Reject $H_0$ | 7.35, 0.00, Reject $H_0$ |

Table 21: Classification accuracies utilizing optimal transport-based embedding for Coil-20 dataset

| | LDA | 1NN | 3NN | 5NN | SVM(Lin) | SVM(RBF) | RF | MLR |
|---|---|---|---|---|---|---|---|---|
| MDS | $0.53 \pm 0.03$ | $0.75 \pm 0.02$ | $0.76 \pm 0.02$ | $0.75 \pm 0.02$ | $0.10 \pm 0.02$ | $0.67 \pm 0.03$ | $0.76 \pm 0.02$ | $0.17 \pm 0.01$ |
| Isomap | $0.51 \pm 0.04$ | $0.72 \pm 0.03$ | $0.68 \pm 0.03$ | $0.67 \pm 0.02$ | $0.31 \pm 0.03$ | $0.53 \pm 0.02$ | $0.72 \pm 0.03$ | $0.30 \pm 0.04$ |
| t-SNE | $0.59 \pm 0.03$ | $0.87 \pm 0.01$ | $0.84 \pm 0.01$ | $0.82 \pm 0.02$ | $0.65 \pm 0.03$ | $0.68 \pm 0.02$ | $0.84 \pm 0.01$ | $0.61 \pm 0.03$ |
| Spec Emb | $0.53 \pm 0.03$ | $0.72 \pm 0.02$ | $0.72 \pm 0.02$ | $0.72 \pm 0.01$ | $0.04 \pm 0.01$ | $0.64 \pm 0.02$ | $0.73 \pm 0.02$ | $0.04 \pm 0.01$ |

Table 22: Classification accuracies utilizing Euclidean-based embedding for Coil-20 dataset

| | LDA | 1NN | 3NN | 5NN | SVM(Lin) | SVM(RBF) | RF | MLR |
|---|---|---|---|---|---|---|---|---|
| MDS | $0.42 \pm 0.03$ | $0.76 \pm 0.02$ | $0.77 \pm 0.02$ | $0.76 \pm 0.02$ | $0.07 \pm 0.03$ | $0.65 \pm 0.03$ | $0.76 \pm 0.02$ | $0.17 \pm 0.05$ |
| Isomap | $0.59 \pm 0.02$ | $0.74 \pm 0.02$ | $0.72 \pm 0.02$ | $0.71 \pm 0.02$ | $0.41 \pm 0.03$ | $0.62 \pm 0.02$ | $0.74 \pm 0.03$ | $0.41 \pm 0.06$ |
| t-SNE | $0.82 \pm 0.02$ | $0.99 \pm 0.01$ | $0.97 \pm 0.01$ | $0.96 \pm 0.01$ | $0.84 \pm 0.02$ | $0.86 \pm 0.01$ | $0.98 \pm 0.01$ | $0.83 \pm 0.02$ |
| Spec Emb | $0.04 \pm 0.01$ | $0.06 \pm 0.01$ | $0.05 \pm 0.01$ | $0.05 \pm 0.01$ | $0.03 \pm 0.01$ | $0.04 \pm 0.01$ | $0.06 \pm 0.01$ | $0.03 \pm 0.01$ |

Table 23: Two-sample t-test comparison results on classification accuracies for optimal transport and Euclidean-based methods for Coil-20 dataset.

| Method | MDS | Isomap | t-SNE | Spec Emb |
|---|---|---|---|---|
| LDA | 7.33, 0.00, Reject $H_0$ | -8.00, 1.00, Fail to Reject $H_0$ | -28.53, 1.00, Fail to Reject $H_0$ | 69.30, 0.00, Reject $H_0$ |
| 1NN | -1.58, 0.94, Fail to Reject $H_0$ | -2.48, 0.99, Fail to Reject $H_0$ | -37.95, 1.00, Fail to Reject $H_0$ | 132.00, 0.00, Reject $H_0$ |
| 3NN | -1.58, 0.94, Fail to Reject $H_0$ | -4.96, 1.00, Fail to Reject $H_0$ | -41.11, 1.00, Fail to Reject $H_0$ | 134.00, 0.00, Reject $H_0$ |
| 5NN | -1.58, 0.94, Fail to Reject $H_0$ | -6.32, 1.00, Fail to Reject $H_0$ | -22.14, 1.00, Fail to Reject $H_0$ | 211.87, 0.00, Reject $H_0$ |
| SVM(Lin) | 3.72, 0.00, Reject $H_0$ | -10.54, 1.00, Fail to Reject $H_0$ | -23.57, 1.00, Fail to Reject $H_0$ | 3.16, 0.00, Reject $H_0$ |
| SVM(RBF) | 2.11, 0.02, Reject $H_0$ | -14.23, 1.00, Fail to Reject $H_0$ | -36.00, 1.00, Fail to Reject $H_0$ | 120.00, 0.00, Reject $H_0$ |
| RF | 0.00, 0.50, Fail to Reject $H_0$ | -2.48, 0.99, Fail to Reject $H_0$ | -44.27, 1.00, Fail to Reject $H_0$ | 134.00, 0.00, Reject $H_0$ |
| MLR | 0.00, 0.50, Fail to Reject $H_0$ | -6.82, 1.00, Fail to Reject $H_0$ | -27.29, 1.00, Fail to Reject $H_0$ | 3.16, 0.00, Reject $H_0$ |

Table 24: Clustering results for MNIST and Fashion MNIST dataset. The top three rows using Euclidean based embedding and the bottom three using optimal transport based embedding.

| Method | Clustering Algorithm | Accuracy | |
|---|---|---|---|
| | | MNIST | Fashion MNIST |
| lsomap | $k$-means | $0.55 \pm 5.51$ | $0.53 \pm 2.80$ |
| | Spectral | $0.14 \pm 0.51$ | $0.13 \pm 0.43$ |
| MDS | $k$-means | $0.55 \pm 3.77$ | $0.52 \pm 3.31$ |
| | Spectral | $0.13 \pm 0.36$ | $0.13 \pm 0.30$ |
| t-SNE | $k$-means | $0.65 \pm 3.96$ | $0.59 \pm 3.52$ |
| | Spectral | $0.55 \pm 5.31$ | $0.53 \pm 0.14$ |
| Wassmap (MDS) | $k$-means | $0.40 \pm 3.00$ | $0.45 \pm 1.63$ |
| | Spectral | $0.54 \pm 2.25$ | $0.35 \pm 2.21$ |
| Wassmap (lsomap) | $k$-means | $0.59 \pm 3.18$ | $0.54 \pm 3.28$ |
| | Spectral | $0.67 \pm 0.48$ | $0.51 \pm 0.64$ |
| Wassmap (t-SNE) | $k$-means | $0.67 \pm 3.79$ | $0.59 \pm 1.56$ |
| | Spectral | $0.65 \pm 3.72$ | $0.57 \pm 2.21$ |

Table 25: Two-sample t-test comparison for clustering between optimal transport-based method and Euclidean-based method for MNIST dataset. Shown are the t statistic and the results of the hypothesis test.

| Method | Clustering Algorithm | t statistic | p values | Results |
|---|---|---|---|---|
| lsomap | $k$-means | 3.14 | 0.00 | Reject $H_0$ |
| | Spectral | 341.97 | 0.00 | Reject $H_0$ |
| MDS | $k$-means | -13.79 | 1.00 | Fail to Reject $H_0$ |
| | Spectral | 80.40 | 0.00 | Reject $H_0$ |
| t-SNE | $k$-means | 1.08 | 0.14 | Fail to Reject $H_0$ |
| | Spectral | 7.51 | 0.00 | Reject $H_0$ |

Table 26: Two-sample t-test comparison for clustering between optimal transport-based method and Euclidean-based method for Fashion MNIST dataset. Shown are the t statistic and the results of the hypothesis test.

| Method | Clustering Algorithm | t statistic | p values | Results |
|--------|---------------------|-------------|----------|---------|
| **lsomap** | $k$-means | 1.03 | 0.15 | Fail to Reject $H_0$ |
| | Spectral | 216.92 | 0.00 | Reject $H_0$ |
| **MDS** | $k$-means | -8.22 | 1.00 | Fail to Reject $H_0$ |
| | Spectral | 43.83 | 0.00 | Reject $H_0$ |
| **t-SNE** | $k$-means | 0.11 | 0.46 | Fail to Reject $H_0$ |
| | Spectral | 8.68 | 0.00 | Reject $H_0$ |

Table 27: Clustering results for 20 Newsgroups dataset using 2 classes (alt. atheism and sci. space) from broader categories (alternative and science), 5 classes ( alt. atheism, comp.sys., mac.hardware, rec.sport.baseball, sci.space, and talk.politics.guns) from 5 different broader categories ( alternative, computer, mac, recreation, science and talk ), and all 20 classes. The top three rows are using the Euclidean based embedding and the bottom three rows are the results using optimal transport based embedding.

| Method | Clustering Algorithm | Accuracy | | |
|--------|---------------------|----------|---|---|
| | | 2 classes | 5 classes | all classes |
| **lsomap** | $k$-means | $0.55 \pm 0.00$ | $0.53 \pm 0.05$ | $0.16 \pm 0.01$ |
| | Spectral | $0.59 \pm 0.00$ | $0.53 \pm 0.01$ | $0.17 \pm 0.00$ |
| **MDS** | $k$-means | $0.55 \pm 0.00$ | $0.24 \pm 0.01$ | $0.10 \pm 0.00$ |
| | Spectral | $0.54 \pm 0.01$ | $0.24 \pm 0.01$ | $0.10 \pm 0.00$ |
| **t-SNE** | $k$-means | $0.61 \pm 0.01$ | $0.60 \pm 0.06$ | $0.26 \pm 0.01$ |
| | Spectral | $0.59 \pm 0.02$ | $0.55 \pm 0.07$ | $0.26 \pm 0.01$ |
| **Wassmap (MDS)** | $k$-means | $0.61 \pm 0.13$ | $0.29 \pm 0.01$ | $0.12 \pm 0.01$ |
| | Spectral | $0.69 \pm 0.06$ | $0.40 \pm 0.02$ | $0.14 \pm 0.01$ |
| **Wassmap (lsomap)** | $k$-means | $0.59 \pm 0.01$ | $0.48 \pm 0.02$ | $0.18 \pm 0.00$ |
| | Spectral | $0.57 \pm 0.00$ | $0.49 \pm 0.00$ | $0.18 \pm 0.00$ |
| **Wassmap (t-SNE)** | $k$-means | $0.59 \pm 0.06$ | $0.46 \pm 0.05$ | $0.17 \pm 0.01$ |
| | Spectral | $0.57 \pm 0.06$ | $0.50 \pm 0.05$ | $0.18 \pm 0.01$ |

Table 28: Two-sample t-test comparison for clustering between optimal transport-based method and Euclidean-based method for Newsgroups data using 2 classes (alt. atheism and sci.space) from the broader categories alternative and science. Shown are the t statistic and the results of the hypothesis test.

| Method | Clustering Algorithm | t statistic | p values | Results |
|--------|---------------------|-------------|----------|---------|
| **lsomap** | $k$-means | 30.90 | 0.00 | Reject $H_0$ |
| | Spectral | -56.67 | 1.00 | Fail to Reject $H_0$ |
| **MDS** | $k$-means | 2.34 | 0.02 | Reject $H_0$ |
| | Spectral | 10.72 | 0.00 | Reject $H_0$ |
| **t-SNE** | $k$-means | -1.35 | 0.9 | Fail to Reject $H_0$ |
| | Spectral | -1.65 | 0.9 | Fail to Reject $H_0$ |

Table 29: Two-sample t-test comparison for clustering between optimal transport-based method and Euclidean-based method for Newsgroups data using 5 classes (alt. atheism, comp.sys., mac.hardware, rec.sport.baseball, sci.space, and talk.politics.guns) from the broader categories alternative, computer, mac, recreation, science and talk. Shown are the t statistic and the results of the hypothesis test.

| Method | Clustering Algorithm | t statistic | p values | Results |
|--------|---------------------|-------------|----------|---------|
| Isomap | $k$-means | -4.5 | 1.00 | Fail to Reject $H_0$ |
|        | Spectral | -22.07 | 1.00 | Fail to Reject $H_0$ |
| MDS | $k$-means | 15.6 | 0.00 | Reject $H_0$ |
|     | Spectral | 37.20 | 0.00 | Reject $H_0$ |
| t-SNE | $k$-means | -8.31 | 1.00 | Fail to Reject $H_0$ |
|       | Spectral | -3.08 | 1.00 | Fail to Reject $H_0$ |

Table 30: Two-sample t-test comparison for clustering between optimal transport-based method and Euclidean-based method for Newsgroups data using all 20 classes. Shown are the t statistic and the results of the hypothesis test.

| Method | Clustering Algorithm | t statistic | p values | Results |
|--------|---------------------|-------------|----------|---------|
| Isomap | $k$-means | 10.04 | 0.00 | Reject $H_0$ |
|        | Spectral | 10.94 | 0.00 | Reject $H_0$ |
| MDS | $k$-means | 18.51 | 0.00 | Reject $H_0$ |
|     | Spectral | 28.75 | 0.00 | Reject $H_0$ |
| t-SNE | $k$-means | -23.48 | 1.00 | Fail to Reject $H_0$ |
|       | Spectral | -26.42 | 1.00 | Fail to Reject $H_0$ |

Table 31: Clustering results Yale Face dataset using 3 classes and 8 classes and all 15 classes. Top three rows for Euclidean based embedding and the bottom three rows for optimal transport based embedding.

| Method | Clustering Algorithm | Accuracy | | |
|--------|---------------------|----------|---|---|
| | | 3 classes | 8 classes | all 15 classes |
| Isomap | $k$-means | $0.70 \pm 0.10$ | $0.62 \pm 0.04$ | $0.49 \pm 0.03$ |
|        | Spectral | $0.88 \pm 0.00$ | $0.58 \pm 0.00$ | $0.57 \pm 0.00$ |
| MDS | $k$-means | $0.72 \pm 0.15$ | $0.63 \pm 0.04$ | $0.56 \pm 0.02$ |
|     | Spectral | $0.81 \pm 0.01$ | $0.69 \pm 0.03$ | $0.64 \pm 0.02$ |
| t-SNE | $k$-means | $0.60 \pm 0.06$ | $0.63 \pm 0.02$ | $0.61 \pm 0.03$ |
|       | Spectral | $0.62 \pm 0.08$ | $0.65 \pm 0.05$ | $0.66 \pm 0.01$ |
| Wassmap (MDS) | $k$-means | $0.43 \pm 0.13$ | $0.48 \pm 0.05$ | $0.27 \pm 0.02$ |
|               | Spectral | $0.85 \pm 0.00$ | $0.59 \pm 0.01$ | $0.56 \pm 0.00$ |
| Wassmap (Isomap) | $k$-means | $0.67 \pm 0.10$ | $0.59 \pm 0.05$ | $0.46 \pm 0.04$ |
|                  | Spectral | $0.88 \pm 0.00$ | $0.72 \pm 0.00$ | $0.60 \pm 0.00$ |
| Wassmap (t-SNE) | $k$-means | $0.46 \pm 0.05$ | $0.60 \pm 0.05$ | $0.59 \pm 0.03$ |
|                 | Spectral | $0.49 \pm 0.08$ | $0.62 \pm 0.02$ | $0.64 \pm 0.02$ |

Table 32: Two-sample t-test comparison for clustering between optimal transport-based method and Euclidean-based method for Yale Face dataset using all 3 classes. Shown are the t statistic and the results of the hypothesis test.

| Method | Clustering Algorithm | t statistic | p values | Results |
|--------|---------------------|-------------|----------|---------|
| Isomap | $k$-means | -0.80 | 0.78 | Fail to Reject $H_0$ |
|        | Spectral | 0.00 | 0.50 | Fail to Reject $H_0$ |
| MDS | $k$-means | -5.10 | 1.00 | Fail to Reject $H_0$ |
|     | Spectral | 8.57 | 0.00 | Reject $H_0$ |
| t-SNE | $k$-means | -5.92 | 1.00 | Fail to Reject $H_0$ |
|       | Spectral | -5.07 | 1.00 | Fail to Reject $H_0$ |

Table 33: Two-sample t-test comparison for clustering between optimal transport-based method and Euclidean-based method for Yale Face dataset using 8 classes. Shown are the t statistic and the results of the hypothesis test.

| Method | Clustering Algorithm | t statistic | p values | Results |
|---|---|---|---|---|
| lsomap | $k$-means | 1.79 | 0.96 | Fail to Reject $H_0$ |
| | Spectral | 3e15 | 0.00 | Reject $H_0$ |
| MDS | $k$-means | -8.31 | 1.00 | Fail to Reject $H_0$ |
| | Spectral | -10.97 | 1.00 | Fail to Reject $H_0$ |
| t-SNE | $k$-means | -11.16 | 1.00 | Fail to Reject $H_0$ |
| | Spectral | -2.62 | 0.99 | Fail to Reject $H_0$ |

Table 34: Two-sample t-test comparison for clustering between optimal transport-based method and Euclidean-based method for Yale Face dataset using all 15 classes. Shown are the t statistic and the results of the hypothesis test.

| Method | Clustering Algorithm | t statistic | p values | Results |
|---|---|---|---|---|
| Isomap | $k$-means | 2.52 | 0.99 | Fail to Reject $H_0$ |
| | Spectral | 44.45 | 0.00 | Reject $H_0$ |
| MDS | $k$-means | -29.26 | 1.00 | Fail to Reject $H_0$ |
| | Spectral | -12.28 | 1.00 | Fail to Reject $H_0$ |
| t-SNE | $k$-means | -1.59 | 0.94 | Fail to Reject $H_0$ |
| | Spectral | -4.27 | 1.00 | Fail to Reject $H_0$ |

