# OpenReview forum: "Optimal transport based dimensionality reduction"
_TMLR — Withdrawn by Authors_

### Review · Reviewer_efiS · 2025-05-16

**Summary Of Contributions:**

This paper studies the use of distance-based machine learning algorithms, when the Euclidean distance is replaced by the Wasserstein metric. The main underlying assumption is that the Wasserstein distance does a better job at capturing the geometry of the data at hand.

**Audience:**

No

**Claims And Evidence:**

No

**Requested Changes:**

As is, the main claim in the paper is a bit misleading (the authors **do not** use OT for dimensionality reduction). On top of this, in my view, this paper does not provide any technical contribution (i.e., no meaningful review of the current literature, no new algorithm), and the experiments are extremely restricted. Given these shortcomings, unfortunately, I don't see any possible changes that would secure my recommendation for acceptance.

**Strengths And Weaknesses:**

__Strengths__

Unfortunately, I don't see any strenghts in this submission.

__Weaknesses__

The main problem with this paper is its main claim and contribution,

> This paper investigates whether modeling image and text data as probability measures and applying optimal transport (OT)-based dimensionality reduction techniques leads to improved performance in downstream machine learning tasks.

which is somewhat misleading. For instance, while there are dimensionality reduction techniques based on optimal transport [1, 2, 3], the authors do not use OT in the design of their dimensionality reduction algorithm. They simply compare data-points under the Wasserstein metric. As a consequence, there is no technical contribution, and the claim that this strategy is **OT-based** is highly disputable.

[1] Van Assel, Hugues, et al. "Snekhorn: Dimension reduction with symmetric entropic affinities." Advances in Neural Information Processing Systems 36 (2023): 44470-44487.

[2] Collas, Antoine, et al. "Entropic Wasserstein component analysis." 2023 IEEE 33rd International Workshop on Machine Learning for Signal Processing (MLSP). IEEE, 2023.

[3] Clark, Ranthony A., Tom Needham, and Thomas Weighill. "Generalized dimension reduction using semi-relaxed Gromov-Wasserstein distance." Proceedings of the AAAI Conference on Artificial Intelligence. Vol. 39. No. 15. 2025.

Besides the latter point, the experiments are designed correctly, but they are small scale compared to most recent work in machine learning literature. For instance, the authors downscale MNIST and Fashion MNIST to only 1000 data points. I assume this is done for being able to compute $O(N^{2})$ Wasserstein distances in a feasible time, but this only shows that the proposed approach is impractical.

---

### Review · Reviewer_RLbG · 2025-05-21

**Summary Of Contributions:**

This paper aims to investigate whether modeling image and text data as probability measures and applying optimal transport (OT)-based dimensionality reduction techniques leads to improved performance in downstream machine learning tasks including classification and clustering. Experiments are conducted using some well-known benchmark datasets.

**Audience:**

No

**Claims And Evidence:**

No

**Requested Changes:**

The authors could thoroughly test OT in LLM-related scenarios and applications to see whether OT benefit LLMs significantly.

**Strengths And Weaknesses:**

The general comment would be an extreme lack of novelty and contribution to the community. All techniques are conventional and the results can be highly predictable. The tested tasks and datasets can be considered just toy experiments.

---

### Review · Reviewer_EiyM · 2025-05-21

**Summary Of Contributions:**

The paper explores an alternative to the standard euclidean distance, namely the Wasserstein distance, for use in dimensionality reduction applications.
The author's modify nonlinear dimensionality reduction techniques (MDS, t-SNE, Isomap) to use the Wasserstein distance and apply the resulting suite of methods to 4 image datasets and a text dataset, then compare the utility of the low-D representations to their euclidean-based counterparts for supervised classification and unsupervised clustering.
Empirical results demonstrate that the benefit of using the Wasserstein metric is dataset specific (and not present at all for some datasets).
The paper concludes with a discussion of the computational complexity of evaluating the Wasserstein distance and a brief discussion of potential strategies for mitigating some of the observed weaknesses of the method.

**Audience:**

No

**Claims And Evidence:**

No

**Requested Changes:**

- For classification and clustering experiments, please add a baseline that applies each technique to the full data space rather than the dimensionality reduced embeddings.
- Change the bolding strategy. In tables 1-4 only the "favorable" results are bolded. It is most typical to bold any instance where one method is statistically significantly superior to alternatives, and this bolding scheme looks at first glance to suggest all significant differences are in favor of Wasserstein embeddings.
- Question: the author's mention that the embedding dimensionality is a critical hyperparamter, and they opt to keep enough dimensions to preserve 95% of the variance. This seems reasonable to me, but I am curious as to which of the Wasserstein or euclidean based methods requires more dimensions to meet this criteria. I'd like to see a table documenting this for each (Metric, Reduction Method) pair.
- I think that the author's need to reconsider the set of datasets they are using to evaluate this technique on. For both text and image classification and clustering, there exist far better techniques than dimensionality reduction --> simple model (i.e. SVM). I am far from an expert on manifold learning or nonlinear dimensionality reduction but I would urge the author's to identify a set of more standard datasets/benchmarks from the literature, along with a more exhaustive set of baselines to compare their method against.

**Strengths And Weaknesses:**

*Strengths*
- The motivation for considering alternative metrics is clearly outlined
- Section 1.3 provides a concise and clear review of distance based nonlinear dimensionality reduction methods
- Empirical experiments are clearly outlined and the procedure is simple to follow

*Weaknesses*
- The exposition on optimal transport was very brief, and could be expanded to make the paper more readable to a wider audience.
- Experiments are limited to toy-scale datasets in both the vision and language domains
- Any advantage of the Wasserstein metric is limited to very simple datasets (i.e. in vision it only has a clear edge on MNIST)
- Applying this methodology to classify such simple datasets seems ill advised. For example, if I were to simply apply an SVM *directly to the pixels* of the MNIST dataset I believe the test accuracy would be similar to if not larger than any of the values in table 1)
- Method is severely more compute expensive than the standard euclidean based approaches, which great limits the scalability in the vision domain.
- The point that the euclidean distance is ill-suited to measuring the distance between images is well taken, but my understanding of methods such as t-SNE is that this is related to the "manifold hypothesis." I.e. because natural images lie on a curved manifold the euclidean distance should only be used locally (and t-SNE seeks an embedding that preserves these distances only at a local scale). I am curious how this view relates to the motivation for finding alternative metrics proposed here.
- Nit: Figures 1/2 should be relegated to appendices, as they do not correspond to any contribution of this work.
- Some of the claims of the conclusion are not well supported (paraphrasing):
   - "The method performs poorly on FashionMNIST because of the presence of visual texture": It would seem to me the increased complexity of the shapes in FashionMNIST could just as well be culpable, and no ablation explicitly supports this claim.
   - "The method is better suited to the text than image domain": The paper only considers one (quite small) text corpus. Is it not possible they would observe similar difficulties if they compared to euclidean distances on larger/more diverse text datasets?

---

### Note · Authors · 2025-06-02

**Comment:**

Thanks to the reviewers for the feedback on the article.

**Withdrawal Confirmation:**

I have read and agree with the venue's withdrawal policy on behalf of myself and my co-authors.